# Fiscal Policy Dilemma in Resolving Agricultural Risks: Evidence from China’s Agricultural Insurance Subsidy Pilot

**DOI:** 10.3390/ijerph18147577

**Published:** 2021-07-16

**Authors:** Yuqiang Gao, Yongkang Shu, Hongjie Cao, Shuting Zhou, Shaobin Shi

**Affiliations:** 1School of Economics, Qingdao University, Qingdao 260071, China; gaoyuqiang209@163.com (Y.G.); 17853500557@163.com (Y.S.); 17806259178@163.com (S.Z.); 2School of Economics, Shandong University, Jinan 250100, China; shishaobin@sdu.edu.cn

**Keywords:** insurance subsidy, agricultural environment, difference-in-difference model, agricultural green development

## Abstract

The agricultural insurance subsidy policy (AISP) encourages farmers to expand production scale by mitigating production risks. Under the high-input production patterns of traditional agriculture, the implementation of AISP is conducive to increase farmers’ income, but it also leads to the destruction of the agricultural environment. Achieving agricultural green development (AGD) has been hindered in China. In this context, this paper attempts to analyze the impact of AISP on farmers’ income and the agricultural environment. Based on the panel data of 316 prefecture-level cities from 2003 to 2012 in China, this paper empirically tests the effects of AISP by employing methods such as time-varying difference-in-difference (DID). The results show that AISP has significantly promoted the growth of farmers’ incomes but has negatively impacted the agricultural environment. Furthermore, the mechanism analysis shows that the policy effects are realized by affecting the quantity of main productive fixed assets (Mpfa) and grain sown area per capita (Gsa). In addition, the policy effect is heterogeneous in different regions. Therefore, the government should appropriately raise the subsidy standard for farmers who adopt environmental-friendly production patterns. At the same time, the government should give more subsidies to the large grain-producing areas.

## 1. Introduction

AGD is a sustainable development plan proposed by China based on its own conditions. AGD requires agriculture to transform from a high-input development patterns to a resource-saving and environmental-friendly development patterns. It is different from agricultural sustainable development, as AGD mainly emphasizes the protection of the agricultural environment. The agricultural environment is the environment on which agricultural organisms rely for survival, development, and reproduction. It mainly includes farmland soil, agricultural water, air, and agricultural organisms. Under the high-input production patterns of traditional agriculture, the agricultural environment is easily affected by production activities. For example, air pollution in agricultural production areas, agricultural water pollution and soil pollution in farmland. Only by realizing AGD can China improve the agricultural environment and promote the high-quality development of agriculture. The realization of AGD depends on the support of national policies, and the government needs to encourage agricultural producers to change production patterns through financial subsidy and other means. Since the reform and opening-up, the Chinese government’s support policies for agriculture have been mostly aimed at stabilizing income and increasing the output of agricultural products. The pollution problem is becoming more and more serious, and achieving AGD has been hampered in China.

China is a country with frequent natural disasters and imperfect agricultural market. In order to mitigate the impact of natural risks and market risks on farmers, the government begins to implement AISP. The policy encourages farmers to participate in insurance by sharing insurance premiums [1]. In the Administrative Measures for the Pilot Program of AISP from the Central Government, the government stipulated that farmers can receive government subsidies when purchasing specific agricultural insurance. At the beginning of the policy, the government mainly included several agricultural products which are planted in the widest range in China and have the greatest impact on people’s lives. Subsidized agricultural products include corn, rice, wheat, soybeans and cotton. In 2007, China officially carried out the pilot reform of AISP in Inner Mongolia, Jilin, Jiangsu, Hunan, Xinjiang, and Sichuan provinces. It was extended to all provinces of China in 2012. AISP does not guide subsidized farmers to adopt environmental-friendly production patterns. With the expansion of production scale, the input of production factors such as pesticides and fertilizers is increasing. The problem of handling agricultural products is also severe. China has achieved agricultural development at the cost of destroying the agricultural environment in a long period of time.

AGD is the common pursuit of many countries, but some countries have failed to achieve AGD. The main reason for this phenomenon is that the subsidy policy is not perfect. The traditional subsidy policy has failed to form a connection with AGD. Therefore, the subsidized farmers did not have the consciousness to change the production patterns and still adopted the high-input production patterns. The conflict between income growth and environmental protection has intensified. Taking China’s AISP as an example, this paper attempts to identify the impact of the subsidy policy without the guidance of the concept of AGD on the agricultural environment and farmers’ income. This will help government departments pay more attention to potential environmental problems when optimizing policies in the future, and formulate subsidy policies that take into account income growth and environmental protection. This paper also attempts to explore the ways in which policies affect the agricultural environment, and points out the role of multiple production factors. The research results can provide suggestions for the government to realize the AGD.

## 2. Literature Review

The research on the income effect and environmental effect shows that AISP has achieved remarkable results. The impact of AISP on farmers’ income is mainly manifested in reducing income fluctuation. Research shows that the agricultural production income of subsidized farmers is more stable after purchasing insurance [2,3]. Agricultural insurance can play a role in hedging risks when the grain yield or the price of grain falls [4] and compensate farmers for losses through a compensation mechanism. AISP increases the expected value of the agricultural production income [5]. Yu [6,7] found that the subsidy policy increases farmers’ income by expanding agricultural planting areas through empirical research. AISP can also play a role in transfer payments. It redistributes national income between the non-agricultural and agricultural sectors in the form of a subsidy [8]. With the support of policies, farmers’ welfare has been improved [9], and agricultural production losses have also been effectively controlled [10]. AISP can also increase farmers’ income indirectly by adjusting the supply and demand of agricultural insurance. The net income that farmers can obtain after risk hedging is declining due to higher insurance premiums. Most farmers do not purchase insurance. Moral hazard and adverse selection are common problems in the insurance business, so the insurance company is unwilling to provide agricultural insurance. AISP can increases the insurance rate by sharing the insurance costs of farmers and granting insurance companies management fee subsidies. Farmers are willing to expand the scale of production after purchasing insurance, so that their income will also increase. Other scholars have researched AISP in Poland [11], China [12], India [13], and the European Union [14]. They have affirmed the role of subsidy policy in promoting farmers’ income. However, some scholars found that the subsidy policy did not positively impact farmers’ income. Take China as an example. Under the current policy of “low insurance premiums, wide coverage, low security, and low remuneration,” crop insurance has no significant impact on farmers’ income [15]. Subsidy policy can also promote the expansion of low-quality farmland, and the risk of a decline in grain yield is becoming increasingly serious [16].

The view that AISP can affect the agricultural environment has been confirmed in many studies. Some scholars believe that AISP is conducive to reducing the nonpoint source pollution [17] and will not cause serious environmental pollution problems [18,19]. Zhong [20] empirically studied the chemical use behavior of farmers after participating in insurance. The results show no significant causal relationship between insurance participation and the amount of chemical use. However, most studies show that the subsidy policy will have adverse effects on the environment. First, financial subsidies encourage farmers to produce in high-risk areas, which will cause damage to the ecological environment. Capitanio [21] conducted an empirical study on the environmental effects of crop insurance subsidies, which was based on the data of 1092 farms in Puglia, southern Italy. The author believes that policy interventions to help farmers cope with risks may have adverse effects on the environment. Walters [22] compared the environmental indicators of different insured areas and found that the insurance plan mainly affects the agricultural environment through the distribution mechanism for land. Second, subsidies have increased the use of chemicals such as pesticides and fertilizers, increasing environmental pollution [23,24]. As the increase in pesticide input can make the output more volatile, it is easier for farmers to obtain insurance compensation [25]. Due to the lack of policy guidance, China’s agriculture has failed to achieve AGD. The pollution problem is particularly prominent after the implementation of AISP. For example, the gas from the disposal of agricultural products can generate air pollution [26,27,28], and the use of chemicals such as pesticides can pollute soil [29,30]. The increasing use of fossil fuels is also an urgent problem in China. With the expansion of production scale, the demand for agricultural machineries has gradually increased, which has promoted the consumption of diesel and other fuels. Most of China’s agricultural machineries do not have exhaust gas treatment equipment, so the excessive use of agricultural machineries has led to a deterioration in air quality. The extensive treatment of straw is another significant cause of agricultural pollution. China’s annual straw production is as high as 700 million tons, and there is no effective treatment technology. In most areas, the traditional open burning method is still used to process straw, bringing massive pressure to the ecological environment.

In terms of research content, the above papers pay more attention to the direct impact of AISP. They mainly empirically tested the change in food production and fertilizer use after the implementation of AISP. Most papers ignored the environmental pollution caused by the treatment of agricultural products, and failed to fully sort out the mechanism of the policy. This paper attempts to make up for these shortcomings. In terms of the research sample, research on China’s AISP is concentrated in a few provinces. This paper selected the panel data of 316 prefecture-level cities, trying to analyze the effect of the policy from the national level. In terms of empirical methods, this paper adopts the time-varying DID method. This method is suitable for analyzing policies that have been implemented in different regions at different times.

## 3. Data and Methods

### 3.1. Data Sources

This paper mainly researches the impact of AISP on farmers’ income and agricultural environment, and attempts to explain how the policy produces effects through mechanism analysis. According to the previous research goals, this paper selected relevant data.

This paper examined data from 316 prefecture-level cities. The per capita net income of rural residents (Pcni) and air quality (PM2.5) were selected as the explained variable to study the impact of AISP. The data of Pcni came from the China Statistical Yearbook For Regional Economy (CSYRE) and the China Rural Statistical Yearbook (CRSY). This paper used the annual average concentration data of PM2.5 retrieved from the atmospheric environment remote sensing image of the National Aeronautics and Space Administration (NASA). The control variables include number of rural households, per capita disaster area, grain yield, fertilizer use per unit area and the level of agricultural modernization, which were calculated as the ratio of the total power of agricultural machinery to the total sown area of crops. The data involved in the control variables came from the CSYRE. In addition, this paper used Mpfs and Gsa to study the policy mechanism. There are many kinds of productive fixed assets, so this paper only selected the representative large and medium tractors as variables. The data for Mpfs and Gsa came from the CSYRE. During the sample period, some prefecture-level cities were merged or split, and a few regions failed to collect statistics on relevant data. Therefore, the data in this paper are unbalanced panel data. At the same time, in order to ensure the accuracy of the research results, this paper specified 2003 as the base period and used CPI to adjust the Pcni. Table 1 shows the descriptive statistics of the main variables in the study.

### 3.2. Research Methods and Model Design

China’s AISP adopts a pilot program followed by promotion. The pilot time varies in different cities to meet the requirements of the time-varying DID model. The specific model is as follows:(1)$$Y_{it}=\alpha +\beta_{1}D_{it}+\beta_{2}Control_{it}+\eta_{t}+\delta_{i}+\varepsilon_{it}$$
where *Y_it_* is the explained variable, including the Pcni or PM2.5. *D_it_* is a policy dummy variable. The value of *D_it_* is 1 after AISP is implemented in city *i*, otherwise the value is 0. The coefficient *β*_1_ is the core indicator to measure the effect of the policy. *Control_it_* represents control variables, *η_t_* is the year fixed effect, *δ_i_* is the city fixed effect, and *ε_it_* is the random error term.

## 4. Results

### 4.1. Benchmark Regression Analysis

The research found that AISP increased the farmers’ income and harmed the air quality. The results are shown in Table 2. In column *a*, the coefficient of *D_it_* was significantly positive, which indicates that AISP has a significant role in promoting the Pcni. Column *b* adds control variables on the basis of column *a*. The regression results show that *β*_1_ was significant at the significance level of 1%. It confirms the income-increasing effect of AISP. Columns *c* and *d* test the causal relationship between AISP and PM2.5. As shown in column *c*, AISP had a negative impact on the PM2.5, but the result was not significant. After adding the control variables, the coefficient of the core explanatory variable was 0.019, and it was significant at the 5% level of significance, indicating that the air quality of each city declined during the pilot period. The policy provides conditions for the farmers’ income growth by mitigating production risks. However, the government’s lack of attention to environmental issues has led to the continuous deterioration of air quality in the pilot cities.

### 4.2. Robustness Test

This paper used three methods to verify the accuracy of the conclusions. The first method was parallel trend testing. The second method was the placebo test. In the third method, the explained variables in Equation (1) were replaced by rural households’ average operating net income and rural households’ average property net income.

#### 4.2.1. Parallel Trend Test

The assumption of a parallel trend is the premise for using the time-varying DID model, requiring that the Pcni in the pilot cities and non-pilot cities had the same trend of change before the implementation of AISP. Therefore, the following model was constructed to carry out a parallel trend test on the sample data. The specifics are as follows:(2)$$Pcni_{it}=\alpha +\beta_{i}\sum_{k=-4}^{4}D^{k}{}_{it}+\tau Control_{it}+\eta_{t}+\delta_{i}+\varepsilon_{it}$$
where *D^k^_it_* is a policy dummy variable of each city based on the policy start year. When the city is in the *k*th year before the pilot, *D^−k^_it_* takes the value 1, and the rest is 0. When the city is in the *k*th year after the pilot, *D^k^_it_* takes the value 1, and the rest is 0. The meaning of the remaining variables remains unchanged. In Figure 1, the estimated results of the coefficients from *D*^−4^*_it_* to *D*^−2^*_it_* indicated that there was no difference in the change trend of Pcni among cities before the introduction of the policy. The reason for the significant coefficient of *D*^−1^*_it_* may be the predictability of the policy, as farmers predicted the direction of the future policy development and responded in advance. Farmers have a lower expected value of production risk when they know that AISP will be implemented in their area. Therefore, farmers expand the scale of production in advance to gain more benefits. Overall, the test results basically met the parallel trend hypothesis.

#### 4.2.2. Placebo Test

Considering the characteristics of the step-by-step pilot of AISP, this paper conducted random non-replacement sampling according to the number of pilot cities in different years. For example, there were 78 pilot cities in 2007, so 78 cities were randomly selected from all 316 cities. This paper assumed that the selected cities began to implement AISP in 2007. Then, the “pseudo” policy dummy variable (*D_it_**) was created to replace *D_it_* in Equation (1). After all sampling was over, a benchmark regression was performed on the new model. At this time, the order of the pilot program was different from the actual one, and AISP should not have had the effect of increasing income. The coefficient (*β*_1_***) of the “pseudo” policy dummy variable should be close to 0. The kernel density figure of *β*_1_*** was drawn after repeating the above operation 1000 times. As shown in Figure 2, the estimated value of *β*_1_*** was evenly distributed around 0. The red line represents the coefficient of the core explanatory variable in the actual test, which was 0.035. Under random sampling, there was no obvious policy effect. This test proved that AISP can only function under the guidance of the government’s reform plan.

#### 4.2.3. Replace the Explained Variable

The purpose of AISP is to decrease production risks and promote agricultural development, so it does not affect the property income of rural families. Therefore, we selected rural households’ average operating net income and rural households’ average property net income as the explained variables and performed the regression test after replacing *Y_it_* in Equation (1). The results are shown in Table 3. In columns *a* and *b*, the core explanatory variable coefficients were 0.039 and 0.044, and both were significant at the 1% significance level, which shows that AISP promotes the operating income of rural families. In columns *c* and *d*, the coefficient of *D_it_* was not significant, regardless of whether the control variable wad added or not, which shows that AISP does not change the property income of rural households. In summary, the policy effect of AISP is unique and does not affect non-agricultural income.

### 4.3. Mechanism Analysis

The implementation of AISP gave farmers an incentive to expand production. At this time, farmers’ investment in production factors such as planting areas and agricultural machinery was conducive to achieving income growth. Large input of production factors also increases the output of agricultural products and increases the use of fossil fuels. The lack of a reasonable treatment will have adverse effects on the agricultural environment. Therefore, Mpfa and Gsa have a major impact on policy effects. To further clarify the mechanism of AISP, this paper built Equation (3) on the basis of Equation (1). The specific model is as follows:(3)$$M_{it}=\alpha +\omega D_{it}+\beta_{2}X_{it}+\eta_{t}+\delta_{i}+\varepsilon_{it}$$
where *M_it_* represents the Mpfa or Gsa. *X_it_* represent control variables, *η_t_* is the year fixed effect, *δ_i_* is the city fixed effect, and *ε_it_* is the random error term.

The research results show that policy can achieve income effects and environmental effects by encouraging farmers to increase Mpfa or Gsa. Table 4 lists the results of the mechanism analysis. In column *a*, the impact of AISP on Mpfa was 0.48, which was significant at the 1% level of significance. AISP increased the rural households’ holding of significant property fixed assets. Further analysis shows that, with the support of AISP, farmers increased their investment in major productive fixed assets, and the mechanization level of production was significantly improved, promoting the increase in the net income of rural residents. At the same time, Mpfa increased the demand for chemical fuel in agriculture. Without effective pollution treatment technology, the air quality has gradually deteriorated. In column *b*, the impact of AISP on Gsa was 0.007, which was significant at the 1% significance level. AISP increased Gsa. Further analysis shows that AISP improved farmers’ income expectations. Under the compensation mechanism of insurance, the expected future income of agricultural production is significantly improved and more stable. Farmers can improve their income level by increasing Gsa. In addition, the development of green agricultural technology in rural areas is uneven, and farmers’ environmental awareness is not strong. Therefore, it is difficult to solve the straw problem caused by the increase in Gsa, and the excessively rapid increase in the amount of straw burned has aggravated air pollution.

### 4.4. Heterogeneity Test

There are obvious regional development differences in China, so the policy effects of different regions may be heterogeneous. To test whether the effects of AISP are heterogeneous, this paper introduced the interaction term (*D_it_* × *Group_i_*) in Equation (1), where *Group_i_* is the grouping variable. This paper used four classification methods. Eastern China generally develops faster than that of other regions, and it has a better economic foundation. This difference may bring about different income effects. Therefore, in the first classification method, the model divided the samples according to geographic location. When the city is in Eastern China, the value of *Group_i_* was 1, and the rest was 0. The jurisdictions of some cities in China contain large grain-producing areas, so they are more susceptible to income effects from AISP. In the second classification method, the value of *Group_i_* was 1 for cities with large grain-producing areas in the jurisdiction and 0 for the rest. China’s annual straw production is unevenly distributed in space, so the environmental effects of AISP may be different. In the third classification method, the sample was classified based on administrative location and straw yield. Among all administrative regions, East China had the largest straw yield, so the value of *Group_i_* was 1 for the prefecture-level cities in East China, and the rest were 0. In the fourth classification method, the paper arranged the prefecture-level cities in ascending order according to the proportion of the primary industry to GDP in 2006. For the top 50% of the prefecture-level cities, *Group_i_* took the value 1, and for the rest it was 0. The purpose was to verify whether the environmental effects of AISP are more obvious in areas where agriculture accounts for a relatively high proportion of the regional economy.

The results of the study are shown in Table 5. Column *a* shows the regression results of the first classification method. AISP had a significant positive impact on the Pcni in each city. On this basis, the coefficient of the interaction term was −0.022, and it was significant at the 10% significance level. That shows that compared with the eastern region, the policy effect in the central and western regions was better. It may be that regions with slow development are more likely to be influenced by AISP. Column *b* shows the regression result of the second classification method. The coefficient of *D_it_* was also significantly positive, and it proves that the income-increasing effect of AISP was universal. On this basis, the coefficient of the interaction term was 0.038, which was significant on the 1% level of significance, indicating that regions with a higher grain output are more likely to increase production and income under the incentives of AISP. Column *c* shows the regression result of the third classification method. The coefficient of *D_it_* was significantly positive, proving the causal relationship between AISP and air pollution. The coefficient of the interaction term was 0.033, which was significant at the 1% level of significance, indicating that areas with large straw production have more serious air pollution during the period of AISP. Column *d* shows the regression result of the fourth classification method. The coefficient of the interaction term was −0.021. On this basis, *β*_1_ was significantly positive, which was significant at the 5% significance level. The areas where the primary industry occupies a large proportion have a larger scale of agricultural production. The yield of straw and the holding of agricultural machineries are much higher than those in other areas. Therefore, air pollution problems in these areas have become more significant after the implementation of AISP.

## 5. Discussion

Based on the practice of China’s AISP, this paper using time-varying DID to identify the policy effect. When researching the income effect of AISP, this paper finds that AISP promoted the income growth of farmers. The research method used in this paper was similar to that of Zhao [15], but the conclusions are quite different. Zhao [15] uses a research method that combines propensity score matching and DID. The author believes that AISP has not significantly increased farmers’ income. The reason for the difference in conclusions may be that the paper [15] only selects the data of farmers in Inner Mongolia as the sample. Different regions have different levels of agricultural development and development patterns. Research on the effects of AISP in only one province may underestimate the actual effects of policy. In the mechanism analysis, this paper selects variables different from Yu [6,7]. Yu [6,7] uses the crop planting area to analyze the policy mechanism. On the basis of the aforementioned article, this paper addes Mpfa to the mechanism analysis. The final conclusions confirm that the subsidy policy can affect the income of farmers through these variables.

By analyzing the environmental effects of AISP, the conclusions drawn in this paper are the same as Capitanio [21]. The government helps farmers mitigate production risks through subsidy policy, which indirectly leads to extensive production and environmental pollution. After the policy is issued, the disposal of agricultural products puts pressure on the agricultural environment. The increase in demand for production factors has also generated air pollution problems. The contradiction between agricultural development and environmental protection has increased. This paper also finds that there are regional differences in the environmental effects of AISP. The empirical results show that areas with higher straw production or a higher proportion of the primary industry in GDP are more significantly affected by AISP. The scale of agricultural production in these areas was high before the implementation of the policy, and the use of agricultural machineries and fossil fuels was much higher than in other areas. Based on policy support, these areas have sufficient foundations to expand production scale, which aggravate environmental pollution. The conclusion that the expansion of production scale can lead to environmental pollution has also been confirmed in the research of Smith [23].

## 6. Conclusions

From the perspective of the environment, the treatment of crop straw needs the government’s attention during the implementation of the policy. In empirical studies, PM2.5 in areas with high straw yields is more susceptible to policy influences. The reason is the lack of technology and facilities for processing straw in rural areas. AISP should not only aim at increasing food production, but also highlight the importance of environmental protection. The government should appropriately raise the subsidy standard for farmers who actively use straw processing equipment or adopt environmental-friendly production patterns. This measure is conducive to the realization of the combination of subsidy policy and environmental protection. The paper also finds that a large amount of investment in agricultural machineries is the cause of environmental pollution. The government needs to set up special machineries purchasing subsidies and promote energy-saving and environmental-friendly agricultural facilities in rural areas.

From the perspective of income, AISP has achieved the goal of income growth by expanding the scale of production. The government should continue to expand the scope of AISP. Most large grain-producing areas are located in central and western China, and local farmers rely on grain production as a source of income. These farmers face greater income risks, and AISP is particularly important to them. The empirical results also showed that the income growth level of farmers in this region is higher than that in the eastern region, so the government should increase the subsidy standards for these areas.

## Figures and Tables

**Figure 1 ijerph-18-07577-f001:**
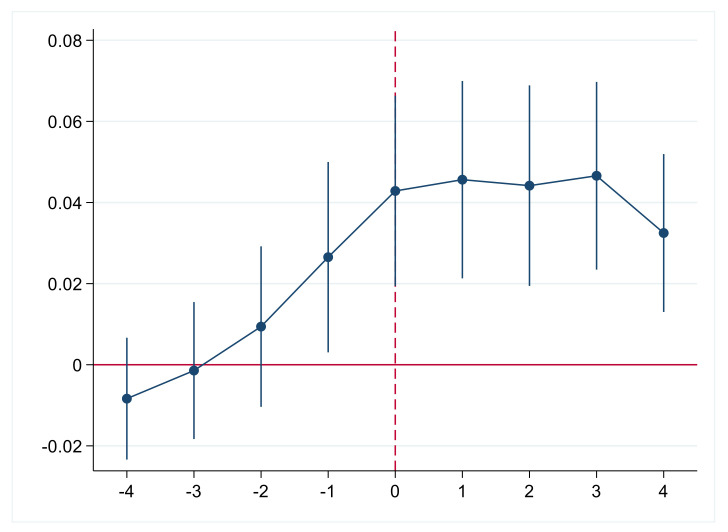
Parallel trend test.

**Figure 2 ijerph-18-07577-f002:**
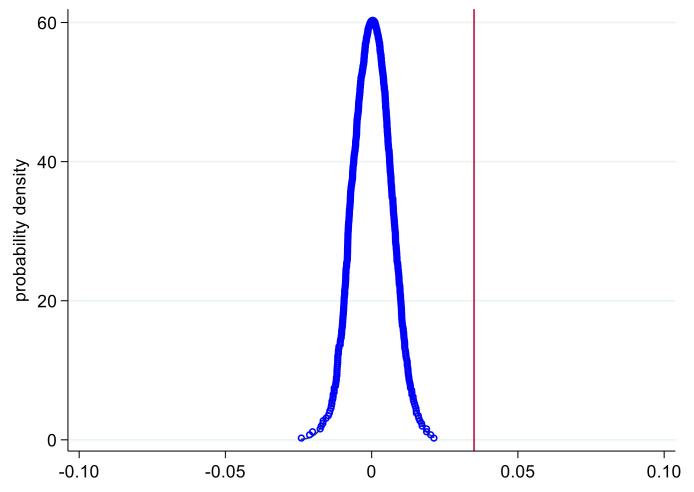
Placebo test.

**Table 1 ijerph-18-07577-t001:** Descriptive statistics of main variables.

Statistic	Variable	Unit	Mean	St.Dev.	Min.	Max.	Observations
Pcni	Per capita net income of rural residents	CNY /per year	8.237	0.471	7.158	9.370	3160
PM2.5	Air quality	mcg/m^3^	3.366	0.634	1.183	4.317	3150
D	Policy dummy variable	-	0.427	0.495	0	1	3160
Lnrh	Number of rural households	Million households	4.051	0.900	1.163	5.469	3053
Pcda	Per capita disaster area	Hectares/person	0.053	0.055	0.006	0.351	3160
Gy	Grain yield	Ten thousand tons	4.681	1.110	0.993	6.702	3019
Fua	Fertilizer use per unit area	Tons/ha	0.332	0.102	0.146	0.546	3160
Lam	The level of agricultural modernization	10 kW/ha	0.512	0.231	0.197	1.180	3160
Mpfa	Quantity of main productive fixed assets	Unit/100 households	2.991	3.302	0.120	17.900	3160
Gsa	Grain sown area per capita	Ha/person	0.128	0.100	0.033	0.605	3160

**Table 2 ijerph-18-07577-t002:** Benchmark regression.

Variable	Pcni		PM2.5	
	(a)	(b)	(c)	(d)
D	0.029 ***	0.035 ***	−0.007	0.019 **
	(0.007)	(0.007)	(0.008)	(0.008)
Lnrh		−0.059 **		0.025
		(0.026)		(0.025)
Pcda		−0.252 ***		0.119 **
		(0.052)		(0.052)
Gy		0.084 ***		−0.074 ***
		(0.023)		(0.021)
Fua		0.754 ***		0.200 **
		(0.176)		(0.086)
Lam		0.106		0.049
		(0.076)		(0.069)
Constant	8.225 ***	7.776 ***	3.369 ***	3.516 ***
	(0.003)	(0.147)	(0.003)	(0.130)
Urban-fixed effect	Control	Control	Control	Control
Year-fixed effect	Control	Control	Control	Control
Observations	3160	2916	3150	2906
R-squared	0.978	0.979	0.979	0.980

Notes: The parentheses are the clustered standard errors at the Prefecture-level city level. *** and ** indicate significant at the 1% and 5% levels, respectively.

**Table 3 ijerph-18-07577-t003:** Robustness checks.

Variable	Operating Income		Property Income	
	(a)	(b)	(c)	(d)
D	0.039 ***	0.044 ***	0.033	0.022
	(0.007)	(0.007)	(0.024)	(0.026)
Lnrh		−0.062 ***		−0.182 ***
		(0.021)		(0.054)
Pcda		−0.195 ***		−0.119
		(0.029)		(0.162)
Gy		0.083 ***		0.115 ***
		(0.019)		(0.032)
Fua		0.524 ***		−0.147
		(0.145)		(0.241)
Lam		0.260 ***		0.474 ***
		(0.090)		(0.159)
Constant	7.700 ***	7.265 ***	4.595 ***	4.609 ***
	(0.003)	(0.133)	(0.010)	(0.267)
Urban-fixed effect	Control	Control	Control	Control
Year-fixed effect	Control	Control	Control	Control
Observations	3159	2915	3159	2915
R-squared	0.968	0.973	0.938	0.941

Notes: The parentheses are the clustered standard errors at the Prefecture-level city level. *** indicate significant at the 1% levels, respectively.

**Table 4 ijerph-18-07577-t004:** Mechanism analysis.

Variable	Mpfa	Gsa
	(a)	(b)
D	0.480 ***	0.007 ***
	(0.109)	(0.001)
Constant	−2.683	0.115 ***
	(2.257)	(0.037)
Control variable	Control	Control
Urban-fixed effect	Control	Control
Year-fixed effect	Control	Control
Observations	2916	2916
R-squared	0.854	0.980

Notes: The parentheses are the clustered standard errors at the Prefecture-level city level. *** indicates significant at the 1% level.

**Table 5 ijerph-18-07577-t005:** Heterogeneity test.

Variable	Pcni		PM2.5	
	(a)	(b)	(c)	(d)
D	0.042 ***	0.022 ***	0.014 *	0.030 ***
	(0.008)	(0.008)	(0.008)	(0.009)
D × Group	−0.022 *	0.038 ***	0.033 ***	−0.021 **
	(0.012)	(0.011)	(0.011)	(0.009)
Lnrh	−0.062 **	−0.057 **	0.023	0.024
	(0.026)	(0.025)	(0.024)	(0.025)
Pcda	−0.265 ***	−0.244 ***	0.116 **	0.109 **
	(0.053)	(0.050)	(0.052)	(0.052)
Gy	0.079 ***	0.075 ***	−0.070 ***	−0.076 ***
	(0.023)	(0.021)	(0.021)	(0.021)
Fua	0.679 ***	0.771 ***	0.186 **	0.180 **
	(0.190)	(0.176)	(0.084)	(0.085)
Lam	0.122	0.103	0.039	0.068
	(0.076)	(0.075)	(0.068)	(0.070)
Constant	7.829 ***	7.808 ***	3.513 ***	3.525 ***
	(0.156)	(0.139)	(0.128)	(0.128)
Urban-fixed effect	Control	Control	Control	Control
Year-fixed effect	Control	Control	Control	Control
Observations	2916	2916	2906	2906
R-squared	0.979	0.980	0.980	0.980

Notes: The parentheses are the clustered standard errors at the Prefecture-level city level. ***, **, and * indicate significant at the 1%, 5%, and 10% levels, respectively.

## Data Availability

Not applicable.

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
