# Peer review of "Fiscal Policy Dilemma in Resolving Agricultural Risks: Evidence from China’s Agricultural Insurance Subsidy Pilot"

_ijerph, 2021, doi:10.3390/ijerph18147577_

Round 1

Reviewer 1 Report

Abstract: should include research background, aim, methodology, data, finding and results

Introduction: does not include specific aim of manustricpt but only general description of assumptions 

Chapter data and results: should also include aim of the manuscript 

Literature review: this chapter is missing 

Discussion: this chapter is missing. In this chapter authors should compare their results with results from previous authors  

I recommend more references from web of science database: Kovacova, M., Kliestik, T., Valaskova, K., Durana, P., Juhaszova, Z. (2019). Systematic review of variables applied in bankruptcy prediction models of Visegrad group countries. Oeconomia Copernicana, 10(4), 743-772. Kliestik, T., Valaskova, K., Nica, E., Kovacova, M., Lazaroiu, G. (2020). Advanced methods of earnings management: Monotonic trends and change-points under spotlight in the Visegrad countries. Oeconomia COpernicana, 11(2), 371-400. Kot, S., Rajiani, I. (2020). Testing and identifying variable dependency through the fisher exact test in central Europe enterprises. Ekonomicko-manazerske spektrum, 14(1), 10-18.Fialova, V., Folvarcna, A. (2020). Default prediction using neural networks for enterprises from the post-soviet country. Ekonomicko-manazerske spektrum, 14(1), 43-51.

Author Response

请参阅附件。

Reviewer 2 Report

The paper deals with the Agricultural Insurance Subsidy Pilot (AISP) in China and analyzes whether the introduction of the AISP has an effect on the income per capita and air pollution at the county-level. The results support the hypnoses of the authors that the induction of AISP helps to increase the farmer’s income and air pollution.

Minor comments

  1. Abbreviations are used before they are defined.
  2. The introduction of the insurance has effects on the income level and the income risk. It would be interesting if the authors could analyze how the income risk is related to the introduction of the AISP. Income risk can be approximated by the time series standard deviation.

Reviewer 3 Report

A review of “Fiscal Policy Dilemma in Resolving Agricultural Risks: Evidence from China’s Agricultural Insurance Subsidy Pilot”

This paper seems to investigate the impact of China’s agricultural insurance subsidy policy on farmers’ income and the environment. It tends to place heavy emphasis on statistical methods to find correlations between the policy and income/environment. The data sources are fine and the findings show that AISP resulted into increasing income and worsening air pollution.

One fundamental question here is this: is this finding on income and pollution significant or does it provide a new insight to what we know already? My answer is negative. The introduction in this paper already shows authors’ assumption that policy incentives can lead to income increase. They seem to have already known this before conducting this research. The conclusion does not mention the significance of the findings in a certain scholarship. If the authors insist that they have a new and significant contribution, they should clarify what it is at least in the conclusion.

Another fundamental question to be asked is this: is this paper somewhat relevant to this journal? I do not think so though the decision may be up to the editor. This journal seems to focus on environmental research and public health. This paper under review tends to have heavy focus on agricultural income and statistical discussion about how a policy incentive may lead to a certain result. Even though this paper touches on air pollution and mentions the “agricultural environment” and “ecological environment,” its argument about policy impact on air pollution is wabbly. Without this paper, it is already clear that agriculture has destroyed the natural environment in the past in various places. I do not yet see a need to conduct various statistical tests to find the connection between agricultural expansion and air pollution. In terms of environmental impact, the authors tend to focus only on PM2.5 whereas there are a few more significant impacts to be examined, including habitat loss (e.g., wetlands, forests), biodiversity, and water pollution (both ground and surface water). In addition, it is not clear what the authors mean by the “agricultural environment.” Simply put, much knowledge acquisition is needed to examine environmental impacts. Finally, is there any connection to public health? Some studies have already shown some connection between biodiversity loss and public health, for example.

Technically, this paper needs to reconsider the title. I do not see what “dilemma” the authors have in mind. Also, this paper does not appear to be discussing about resolving agricultural risks.

Reviewer 4 Report

Summary and overall opinion

The manuscript deals with the impact of agricultural insurance subsidy policies. The theoretical construction is interesting although the research questions are questionably treated. The discussion is generally coherent but it also tends to be vague at times and the manuscript seems updated in terms of references. The manuscript has a clear dose of intrinsic merit but is not publishable as it stands.

Comments and Suggestions 

2.1. Main concerns

1. First of all, at first glance the manuscript does not fall under the scope of the journal. 

2. The merit of the manuscript resides in its methodological exercise, however, past this point certain aspects raise concerning question marks. 

3. The introduction is perhaps the weakest section of the manuscript as it is lightly constructed. The following problems must be addressed more substantially. 

a. The authors must upgrade their motivation for the research question.

b. The gap in the literature needs to be highlighted in a more relevant manner.

c. Most importantly, the authors should offer a clear discussion on the original contributions of the manuscript. The paragraph put forward in the current version does very little in this direction. 

4. The manuscript would benefit from a literature review section. 

5. Although the method employed is adequate, the manuscript reports results for the 2003 - 2012 range. Although there is atomicity in the data employed, I would like to see the opinion of the authors about the impact to general knowledge brought by an investigation focusing on that specific time frame given the recent global dynamics of the agrifood sector. 

6. Section 4 is poorly constructed and written too much from the perspective of the method employs. I would like to see a total upgrade of the section with consistent efforts in terms of drafting.

7. The manuscript requires a consistent discussion section built to complement the result section.

8. In the conclusions section I would like to see several clear takeaways that derive from the analysis. 

2.2. Minor concerns

1. Comments on variable choice would be a plus.

2. The article needs a consistent revision in terms of language and grammar. 

All in all, before being reconsidered the manuscript requires in my opinion a very heavy dose of upgrades in drafting. This applies to almost all sections of the manuscript and will result in a dramatic change of the original version

Round 2

Reviewer 1 Report

Authors accepted recommendations 

Author Response

Thank you very much for your valuable comments and recognition of our revised contents. We have done our best to revise the manuscript, but if any additional revision is needed, we will certainly do so under your directions.

Reviewer 3 Report

Review of “Fiscal Policy Dilemma in Resolving Agricultural Risks”

I see some improvement from the last draft. The authors now appear to show some connection between insurance subsidy and environmental impact. In essence, they seem to argue that AISP led to air and soil pollution, especially in some regions with more straw production and low-income households. However, the authors appear to be still confused about what they are actually claiming in terms of findings and contribution to scholarship. Also, added sentences show hasty and careless use of terms and grammar problems. In addition to speediness, I encourage the authors to pay more attention to quality. I give more detailed comments below.

Abstract

The authors use the “agricultural environment” and “environmental governance” interchangeably. They say that AISP does not consider environmental governance. This loose terminology use is confusing without clear definition of these terms at least in the Introduction or the rest of the main discussion.

This abstract makes it difficult to understand the main focus of this paper: is this about the impact of AISP on the agricultural environment? Or is this about the impact of AISP on environmental governance? Also, after reading the introduction, I wonder why the authors do not mention anthing about agricultural green development, which they emphasize in the Introduction. Careful use of terms is essential to clarify the scope and focus of the paper. Keywords include only “environment,” but not environmental governance, agricultural environment, agricultural green development.

The last two sentences here need to reflect on revised discussion in the conclusion section. Explain what they mean by improving “the quality of subsidies to achieve the green development of agriculture.” Do they mean to target certain types of crop production (e.g., rice, wheat) or low-income households? What do “different subsidy programs” mean, given that AISP does not appear to have improved environmental sustainability? If AISP does not consider environmental governance, what “different subsidies” can fill this policy gap? Does the Chinese national or provincial government have any subsidy programs to promote green development in connection to agriculture?

Introduction

The first paragraph, which was added in this revised paper, needs citation to support their argument. So far, this paragraph sounds like a policy propaganda by the Chinese government. Also, explain if there are any differences between AGD and sustainable agriculture. What are trends and characteristics of past research on AGD or sustainable agriculture? Considering these, what does this paper offer to our understanding of AGD?

The last paragraph, which was also added, emphasizes the usefulness of this paper for Chinese policymaking in the future. Setting aside the soundness of this claim, the authors also need to help international academic readers understand what this paper implies to better understanding a certain aspect of academic questions. Do we learn about studies on “insurance, agricultural risk, DID model or environment”? (See your keywords.)

A few terminology definitions are needed to improve discussion here: “agriculture” in connection to AISP (what activities do AISP cover?); “agricultural environment” in connection/contrast to the natural environment; “environmental governance” in connection to the agricultural environment.

Literature review

The first paragraph (p.2) emphasizes the positive direct/indirect impacts of AISP on farmer’s income. Is there any connection between farmer’s income and environmental governance? In other word, do farmers with increased income care more about sustainability or greener practices (somewhat in reminiscent of the Environmental Kuznets Curve)? If not, how do you connect subsidy, income and environmental governance?

The last paragraph emphasizes a lack of “comprehensive” study on policy impacts of AISP. After reading this paper, it does not seem to provide any glimpse of a “comprehensive” insight. Do they mean by nation-wide research? Do they discuss all forms of crop production? Or do they focus on rice production? (After reading the discussion on regional differences in response to AISP [lines 366-367], this paper seems to be focusing on rice or wheat.)

Reviewer 4 Report

First of all, I would like to congratulate the authors for their efforts put into providing a more refined version of the manuscript. The authors made a fair effort in tackling some of the concerns I expressed regarding the initial version of the paper. My present review will follow in the line of the comments put forward in my original review, trying to treat them in the same order.
Comment 1 is partially discussed. In my opinion, it still stands.
Comment 2 has been addressed.
Comment 3 has been partially addressed and I would have expected more work to be done in this direction.
Comment 4 has been dealt with.
I understand the arguments of the authors brought for Comment 5.
I understand the arguments of the authors brought for Comment 6 and I consider them to be fair.
Comments 7,8, 9, and 10 have been answered.
